# A family of fluoride-specific ion channels with dual-topology architecture

**Randy B Stockbridge, Janice L Robertson[†], Ludmila Kolmakova-Partensky, Christopher Miller\***

Department of Biochemistry, Howard Hughes Medical Institute, Brandeis University, Waltham, United States

**Abstract** Fluoride ion, ubiquitous in soil, water, and marine environments, is a chronic threat to microorganisms. Many prokaryotes, archea, unicellular eukaryotes, and plants use a recently discovered family of $F^-$ exporter proteins to lower cytoplasmic $F^-$ levels to counteract the anion's toxicity. We show here that these 'Fluc' proteins, purified and reconstituted in liposomes and planar phospholipid bilayers, form constitutively open anion channels with extreme selectivity for $F^-$ over $Cl^-$. The active channel is a dimer of identical or homologous subunits arranged in antiparallel transmembrane orientation. This dual-topology assembly has not previously been seen in ion channels but is known in multidrug transporters of the SMR family, and is suggestive of an evolutionary antecedent of the inverted repeats found within the subunits of many membrane transport proteins.

## Introduction

Fluoride pervades our biosphere, appearing in groundwater, sea, and soil typically at 10–100 µM levels (*Weinstein and Davison, 2004*). This ubiquitous inorganic xenobiotic inhibits two enzymes essential for glycolytic metabolism and nucleic acid synthesis: enolase and pyrophosphatase (*Marquis et al., 2003*; *Samygina et al., 2007*). Accordingly, many unicellular organisms, as well as green plants, use $F^-$ exporter proteins in their cell membranes to keep cytoplasmic $F^-$ concentration low, thereby minimizing the anion's toxic effects (*Baker et al., 2012*; *Stockbridge et al., 2012*). Two separate, phylogenetically unrelated families of $F^-$ exporters are now known: $CLC^F$-type $F^-/H^+$ antiporters, a subclass of the widespread CLC superfamily of anion-transport proteins, and a group of small membrane proteins known as the 'crcB' family or, as we rename them here (to avoid confusion with the CLCs), the 'Fluc' family. Fluc proteins are widespread among unicellular organisms. Deletion of the single Fluc gene in *Escherichia coli* produces hypersensitivity to $F^-$, and this growth-phenotype may be rescued by $F^-$ exporter genes from a variety of bacterial species (*Baker et al., 2012*; *Stockbridge et al., 2012*). By expressing, purifying, and functionally reconstituting several bacterial Fluc homologs, we demonstrate that these are highly selective $F^-$-conducting ion channels constructed as dimers of identical or homologous membrane-embedded domains arranged in an inverted-topology fashion. This type of molecular architecture is unprecedented among ion channels but is reminiscent of the dual-topology construction of small multidrug transporters (*Rapp et al., 2006*; *Schuldiner, 2009*; *Morrison et al., 2012*) and of the inverted structural repeats appearing in many membrane transport proteins (*Forrest et al., 2010*).

## Results

Prokaryotic Fluc sequences consist of 120–130 residues with four predicted transmembrane (TM) helix motifs (*Figure 1A*), short N- and C-termini, and a strongly conserved sequence (GxxGGxTTFSTFxxE) in TM3. A survey of 47 bacterial Flucs yielded several biochemically and electrophysiologically tractable homologues, nicknamed here Ec2, Bpe, La1, La2 (*Table 1*); we also examine several fused constructs

**\*For correspondence:** cmiller@brandeis.edu

**†Present address:** Department of Molecular Physiology and Biophysics, University of Iowa, Iowa City, United States

**Competing interests:** The authors declare that no competing interests exist.

**eLife digest** Fluorine is the thirteenth-most abundant element in the Earth's crust, and fluoride ions are found in both soil and water, where they accumulate through the weathering of rocks or from industrial pollution. However, high levels of fluoride ions can inhibit two processes essential to life: the production of energy by glycolysis and the synthesis of DNA and RNA bases. In polluted areas, organisms such as bacteria, algae and plants must remove fluoride ions from their cells in order to survive.

Since ions cannot freely cross lipid membranes, organisms use proteins called channels or carriers to move ions into and out of their cells. Channel proteins form a pore, or channel, in the cell membrane, through which ions can quickly move from areas of high concentration to areas of low concentration. In contrast, carrier proteins can transport ions in both directions—that is, to and from areas of high concentration—but they are slower than channel proteins.

A family of proteins that export fluoride from microbe and plant cells, thus allowing them to grow in the presence of this toxic ion, was discovered recently, but it was not clear if these proteins function as channels or as carrier proteins. Now, Stockbridge et al. find that these proteins, called Fluc proteins, are fluoride channels with an unusual architecture.

Fluc proteins are found in many species of bacteria, and Stockbridge et al. show that a number of these, when purified and inserted into a lipid membrane, are channel proteins. Additionally, they do not transport related ions such as chloride, which means that they are unusually selective for ion channels. Two Fluc polypeptides associate to form a channel in the cell membrane, and Stockbridge et al. show that these two subunits are arranged in an antiparallel formation. Although this architecture is unprecedented among ion channels, it has been observed in carrier proteins in a range of organisms, and may indicate that Fluc proteins offer an evolutionary model for many carrier proteins.

in which La1 and La2 are joined together in tandem into single polypeptides. The purified proteins run cleanly on SDS-PAGE (*Figure 1—figure supplement 1*) near the predicted molecular weights and migrate on size-exclusion columns (SEC) as monodisperse peaks about 0.6 ml ahead of the expected monomer position (*Figure 1—figure supplement 2*). Both Ec2 and Bpe are particularly stable, with chromatographic profiles unaltered even after several days in detergent solution at 37°C.

## Fluc is a four-TM membrane protein

The predicted transmembrane topology of Fluc (*Figure 1B*) was experimentally tested with Bpe, a naturally cysteine-free homologue poised for specific labeling at a unique cysteine substituted near the N-terminus (T3C). Lipid vesicles reconstituted with this mutant were treated with LysC protease to exclusively remove the C-terminal His tags exposed to the outside of the liposomes. *Figure 1C* shows that Fluc proteins are randomly oriented in the liposomes. About half of the protein population has an externally exposed C-terminus and is susceptible to cleavage, as indicated by faster migration on SDS-PAGE. The other half is protected, with the C-termini exposed to the protease-inaccessible liposome interior.

After proteolysis, a membrane-impermeant, thiol-reactive fluorophore was added to the liposome suspension to label externally exposed N-termini. This treatment labels only the lower band containing the polypeptides with externally exposed C-termini, thereby demonstrating that the N- and C-termini are exposed on the same side of the membrane, as anticipated by the Fluc hydropathy profile. To further confirm the predicted four-TM topology, similar experiments were performed with unique cysteines (N31C and E94C) placed in the first or third loops (*Figure 1C*). Here, the label reacts exclusively with the proteins that retain the C-terminal His-tag, thus placing these loops and the C-terminus on opposite sides of the membrane.

## Fluc proteins form F⁻ specific ion channels

Purified Fluc proteins are functional, as seen in F⁻ and Cl⁻ efflux experiments in reconstituted liposomes (*Stockbridge et al., 2012*), with ion-specific electrodes following the appearance of the anions in the external solution. Liposomes were reconstituted under 'Poisson-dilution' conditions, that is, with Ec2 or Bpe at protein densities so low that 30–50% of the liposomes are devoid of protein, and most of the protein-containing liposomes carry only a single functional unit (*Walden et al., 2007*). The liposomes

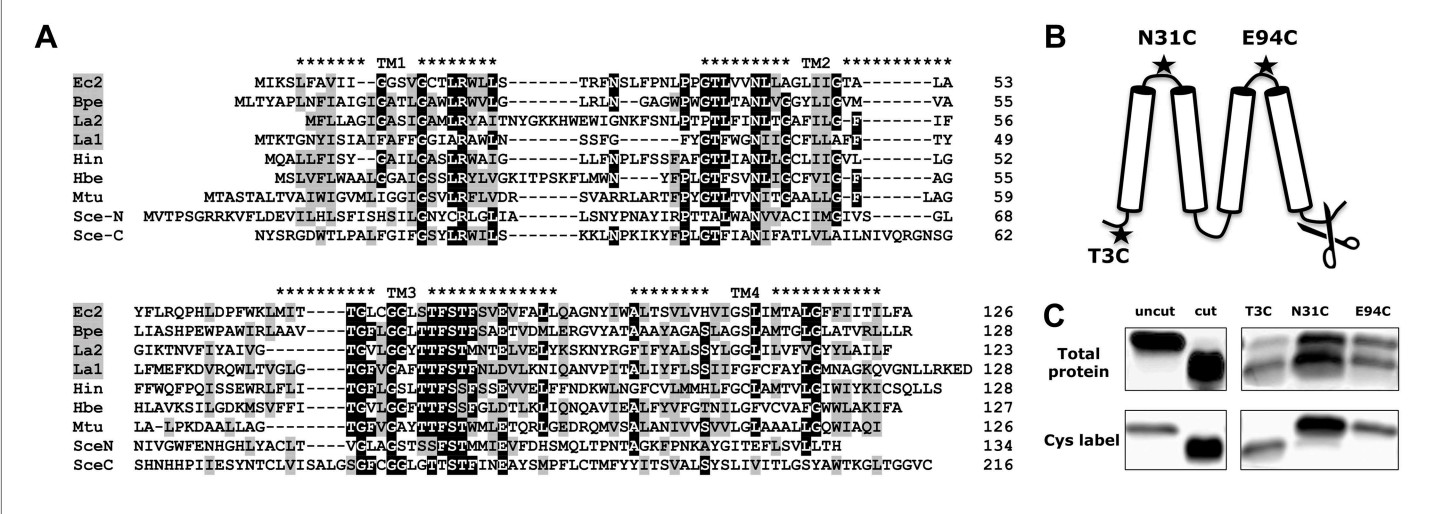

**Figure 1**. Fluc sequence and topology. (**A**) Alignment of Fluc homologues Ec2, Bpe, La1, and La2 (shaded, this study), and from *Haemophilus influenza* (Hin), *Helicobacter bezzozeronii* (Hbe), *Mycobacterium tuberculosis* (Mtu) and the N- and C-terminal domains of *Saccharomyces cerevisiae* (Sce), excluding 75 residues of the N-C linker. Expressed Flucs shown on 15% SDS-PAGE gel of purified preparations (**Figure 1—figure supplement 1**) and size-exclusion chromatograms after the Co-affinity column step (**Figure 1—figure supplement 2**). (**B**) Model of 4-TM Fluc subunit with positions of unique cysteines and cleavable His tag indicated. (**C**) SDS-PAGE of Fluc-Bpe variants with unique cysteines indicated. Left panels: Bpe (T3C) before and after complete proteolysis in detergent to remove the C-terminal His tag followed by cysteine-specific labeling with Alexa647. Right panels: same treatments of proteins reconstituted in POPC/POPG liposomes. Upper panels: total protein stained with amino-reactive fluorescein. Lower panels: protein stained with cysteine-conjugated Alexa647.

The following figure supplements are available for figure 1:

**Figure supplement 1**. Purified Fluc proteins Ec2, Bpe, and Laf-TM as indicated.

**Figure supplement 2**. Size exclusion chromatograms (Superdex 200) of proteins used in this study.

were loaded with 150 mM each of KF and KCl and suspended in iso-osmotic solution containing 1 mM KF and KCl along with 300 mM K-isethionate (2-hydroxyethanesulfonate, a membrane-impermeant monovalent anion). Valinomycin (Vln), a $K^+$ ionophore, was then added to start the efflux by allowing the anions to move down their gradients. After protein-catalyzed ion efflux had reached completion, detergent was added to disrupt all the liposomes and reveal the protein-free fraction. Two striking results are apparent from the $F^-$ and $Cl^-$ efflux kinetics (**Figure 2A**). First, the proteoliposomes are selectively permeable to $F^-$; $Cl^-$ efflux is undetectable on this timescale, and similar experiments in the absence of $F^-$ recapitulate this lack of $Cl^-$ transport. Second, Fluc-mediated $F^-$ efflux is so fast that an initial rate measurement is precluded by the 1-s dead-time of the stirred-cuvette system. An estimate based on a 35-nm liposome radius leads to a lower limit of ~30,000 $s^{-1}$ for the single-Fluc transport

**Table 1.** Fluc constructs used in this work

| Construct | Source | NCBI ref. seq. | Linker | Yield, µg/l |
|---|---|---|---|---|
| Ec2 | *E. coli* virulence plasmid | YP_001481330.1 | – | 120 |
| Bpe | *Bordatella pertussis* | NP_879990.1 | – | 50 |
| La1 | *Lactococcus acidophilus* | YP_193874.1 | – | 150 |
| La2 | *Lactococcus acidophilus* | YP_193873.1 | – | 5 |
| Laf-TM | La1-La2 fusion | – | GTGAAGASRSLERRITLILFGVMAL VIGTILLLLYGIGSGACGAS | 50 |
| LapA | La1-La2 fusion | – | GTTRGKAASLVPAS | 20 |
| LapB | La1-La2 fusion | – | GTEFEAYVEQKLISEEDLNSAVDAAS | 20 |

The underlined sequence in the Laf-TM linker indicates the glycophorin A transmembrane helix.

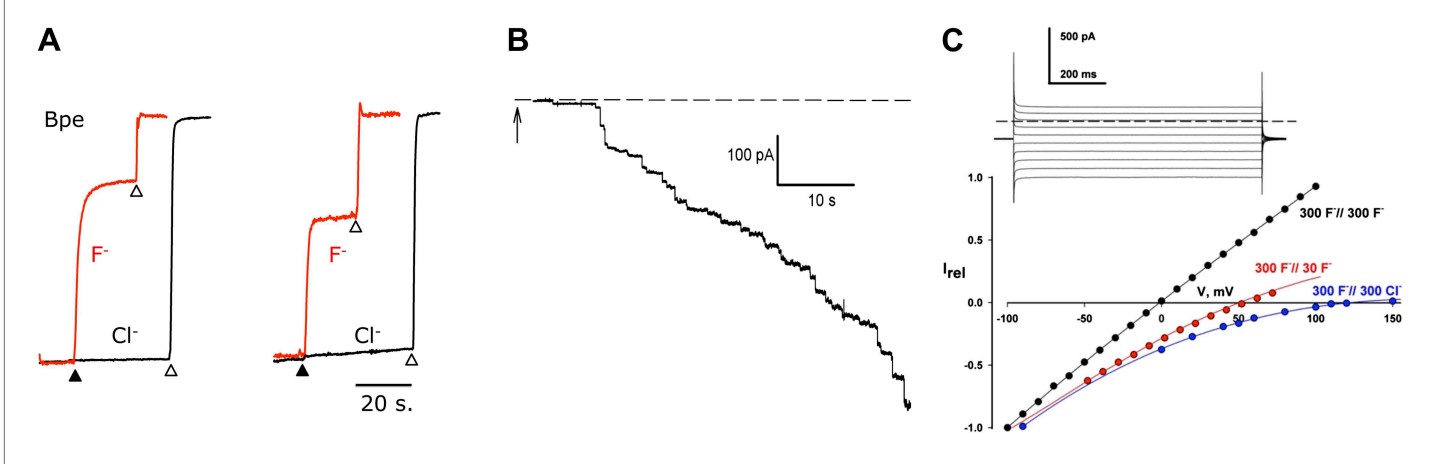

**Figure 2**. F⁻ transport and selectivity in Fluc proteins. (**A**) Liposome flux assays for Fluc Ec2 and Bpe (~10 pmol protein/mg lipid). Anion efflux from liposomes containing KF+ KCl, 150 mM each, was initiated by addition of 1 µM Vln (filled triangle). Anions trapped in protein-free liposomes were released with 50 mM β-OG (open triangles). Anion appearance in the external solution was monitored (F⁻ red, Cl⁻ black traces), and signals normalized to final levels. (**B**) Timecourse of insertion of Fluc-Ec2 reconstituted liposomes (5 µg/mg) into a planar bilayer under salt-gradient conditions (300 mM NaF/30 mM NaF) at −100 mV. Arrow indicates addition of 0.5-µl liposomes. Dashed line is zero-current level. (**C**) Macroscopic I–V relations under ionic conditions indicated. Inset: Current responses to voltage pulses from −90 mV to +90 mV in 20-mV increments under salt-gradient conditions as in (**B**). Reversal potentials from 4–5 separate bilayers were 53 ± 1 mV and 119 ± 3 mV for salt-gradient and bi-ionic conditions, respectively.

rate. Since this is substantially higher than turnover of any conformational-cycle based membrane transporter (*Jayaram et al., 2008*), it intimates that Fluc might be a F⁻ channel.

Accordingly, we recorded ionic currents mediated by Fluc-Ec2 inserted into planar phospholipid bilayers. Initial experiments aimed at maximizing the electrical signal used liposomes with high protein density, 10–50 Fluc copies per liposome. When fused into planar bilayers, these liposomes evoke a robust increase in bilayer conductance in discrete steps (*Figure 2B*), each of which reports a single liposome inserting its packet of ion-conducting proteins. The heterogeneity in step-size reflects the wide distribution of liposome size (*Miller and Racker, 1976*; *Accardi et al., 2004*). After a few minutes, liposomes are perfused away to stabilize the bilayer conductance, thereby allowing examination of its properties. We applied a series of pulses to voltages from −100 mV to +150 mV and recorded the resulting currents to determine current-voltage (I–V) relations under various ionic conditions. As shown in the inset to *Figure 2C*, recorded in the presence of a salt gradient (300 mM NaF//30 mM NaF), the current appears as a 'leak' without any time- or voltage-dependent gating over a wide voltage range. The Fluc-mediated conductance is ideally selective for F⁻ over Na⁺, as seen from the zero-current voltage (reversal potential) of 53 mV, indistinguishable from the Nernst potential for F⁻ in the eightfold ionic activity gradient. High selectivity is further illustrated by the I–V relation under bi-ionic conditions (300 mM NaF//300 mM NaCl), whose high reversal potential (119 mV) sets a lower limit of ~100 on the F⁻/Cl⁻ permeability ratio.

To detect single-channel currents, we prepared Fluc-reconstituted liposomes under Poisson-dilution conditions and fused them as above into planar bilayers. *Figure 3A* shows examples of Fluc-insertion events in three separate bilayers, along with a histogram of Fluc insertion-step amplitudes taken from ~50 bilayers (*Figure 3B*, upper panel). Most of these insertion-steps are 1.8 pA under these conditions, equivalent to 7 pS, while a minority (<20%) are 30–50% of this main step-size (illustrated in the third trace). The current shows occasional discrete fluctuations of two types: subsecond-timescale closing events and rare millisecond-timescale excursions to a substate of 50–60% of the main-state amplitude. The 'full-open' state persists >95% of the time at all voltages (−200 mV to +200 mV). Its high open probability further shows itself in an extended recording with three channels in the bilayer (*Figure 3A*, red trace) and from the amplitude histogram of a typical single channel (right panel lower histogram). These channels display voltage-independent, anion-selective characteristics in harmony with the macroscopic currents reported above, and their appearance is strictly dependent on reconstituting Fluc protein into the liposomes. Accordingly, we conclude that these unitary current-steps reflect activity of single Fluc proteins, with single-molecule turnover of ~10⁷ F⁻ ions/sec at −200 mV, a rate

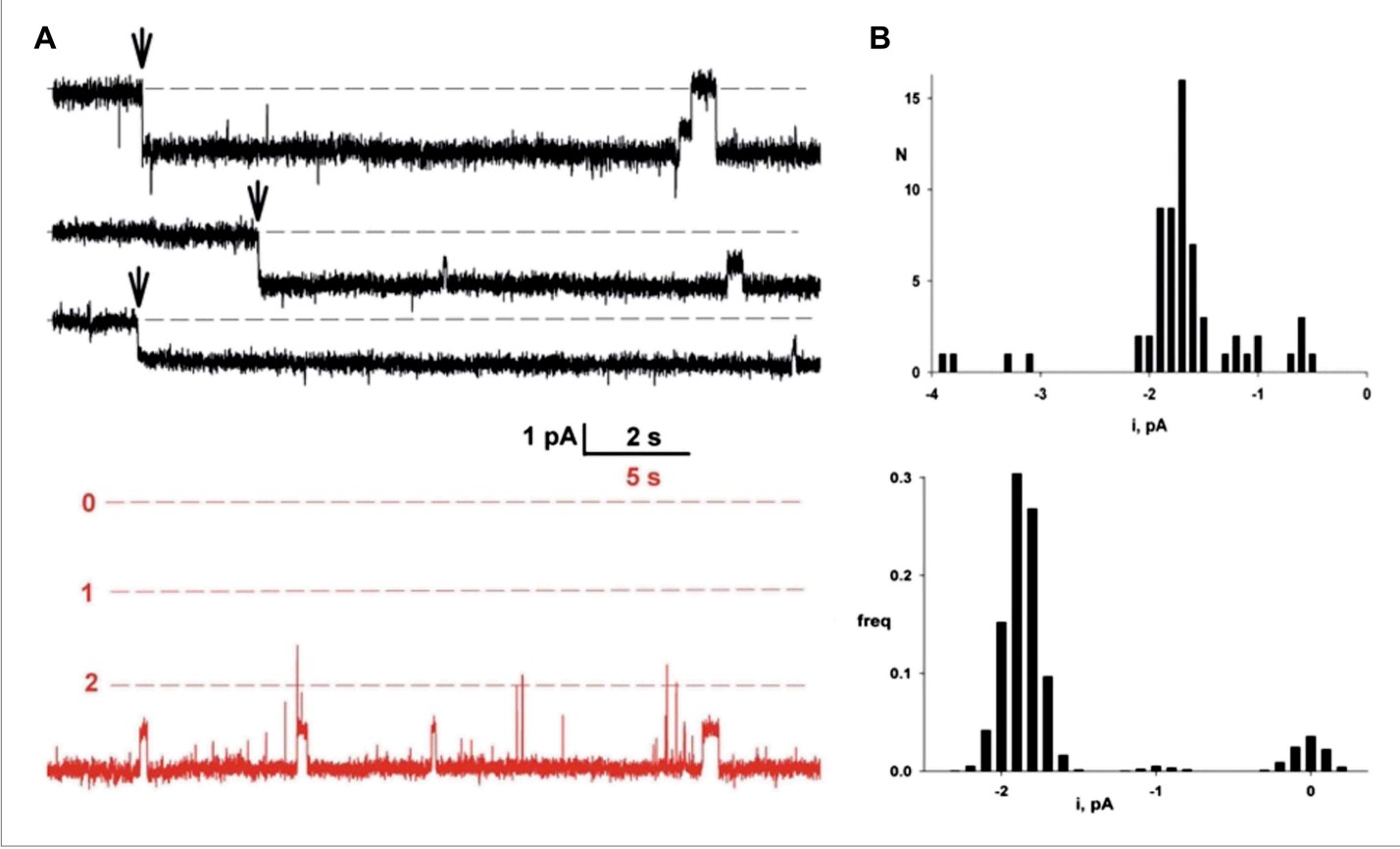

**Figure 3**. Single Fluc channels. Fluc-Ec2 reconstituted liposomes (0.1 µg/mg) were inserted into planar bilayers under salt-gradient conditions, and current fluctuations (downward opening) were recorded at −200 mV. (**A**) Upper traces: three examples of the first Fluc insertion events (arrows) after adding liposomes to the bilayer. The two top traces are representative of most of the Fluc insertions, while the third trace illustrates the rarer low-conductance events. Lower trace in red is taken from a bilayer containing three Fluc channels (open levels marked with dashed lines). (**B**) Upper panel: first-insertion current histograms for 63 separate insertion events taken from ~50 bilayers. Lower panel: all-points amplitude histogram for a single Fluc channel (lower panel) recorded for ~20 s. In both histograms, zero current is defined as the mean level of the fully closed channel.

typical for ion channels and three to four orders of magnitude higher than the fastest known transporters. Thus, Fluc-Ec2 is a highly $F^-$-selective channel, a protein with a transmembrane pore through which $F^-$ ion moves thermodynamically downhill by electrodiffusion.

## Fluc channels are dimers

Fluc-Ec2 runs as a homodimer in detergent micelles on a size-exclusion column, as determined by combined measurements of UV absorbance, refractive index, and static light scattering of the eluting peak (***Figure 4A***). Since an ion channel pore formed on a dimer axis is unprecedented, we also assessed Fluc's oligomeric state in its native habitat, the lipid bilayer. For this, we prepared Fluc-Bpe for single-molecule total internal reflection fluorescence (TIRF) photobleaching experiments (***Tombola et al., 2008***) by labeling a unique cysteine mutant, R29C, with Cy5-maleimide and reconstituting the protein at a very low density to favor single-channel liposomes. Upon illumination of these liposomes immobilized on a glass surface, the covalently attached fluorophores bleach in several minutes (***Figure 4B,C***), mostly in single and double steps (***Figure 4D***). Given the measured labeling efficiency (72%), the preponderance of single- and double-bleach events is fully consistent with a dimer (***Figure 4E***). Fewer than 5% bleach in three or four steps, consistent with co-localization of the liposomes, but not with trimeric or tetrameric architecture for the channel. The same conclusion was obtained in a second dataset using a different Bpe mutant, T3C (***Figure 4—figure supplement 1***).

We further tested the oligomeric state of Fluc in membranes by exploiting the $F^-$-transport function of the channel in 'Poisson-dump' experiments. The crux of this approach is the Poisson statistics of

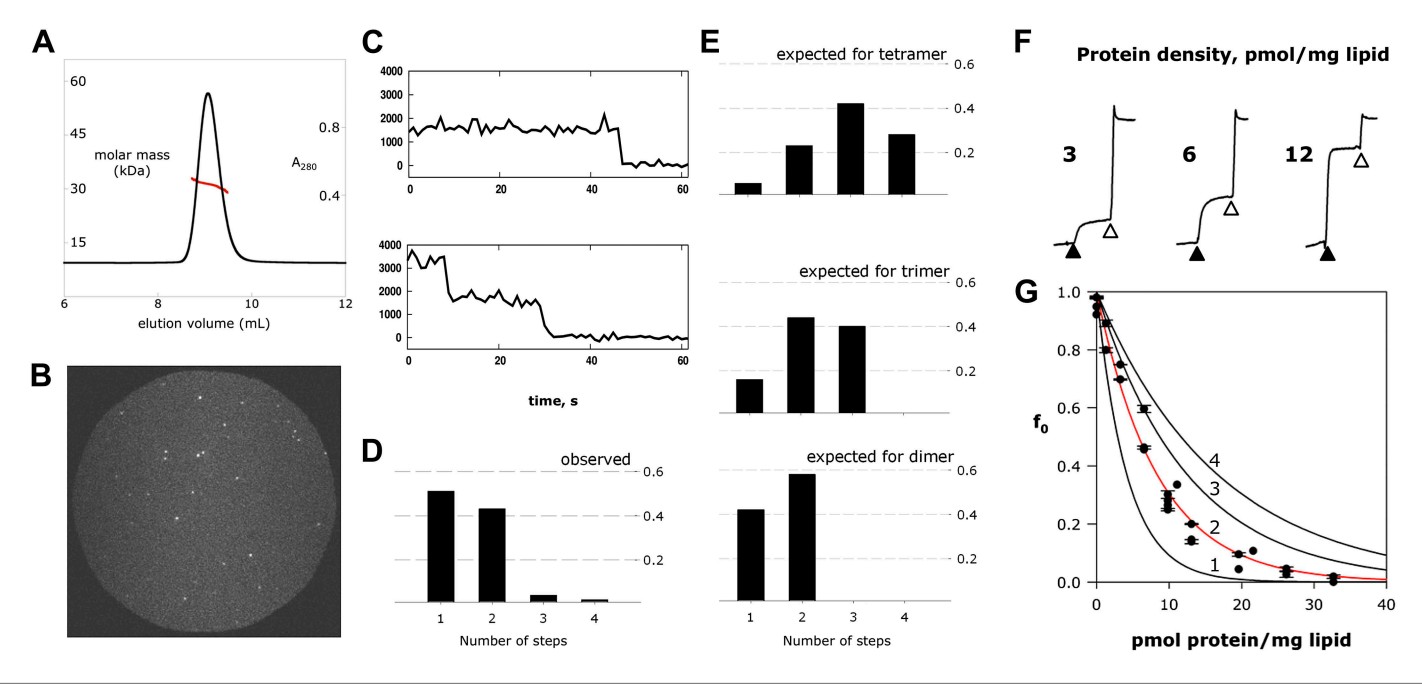

**Figure 4**. Fluc homodimer in micelles and membranes. (**A**) SEC-SLS/UV/RI elution profile for Fluc-Ec2 with the molar mass of the eluting peak calculated using the ASTRA technique. (**B**) TIRF field of immobilized liposomes containing Bpe-R29C labeled with Cy5-maleimide. (**C**) Representative one- and two-step photobleaching events, integrated intensity in arbitrary units. (**D**) Observed populations of 1-, 2-, 3-, and 4-step photobleaching for Bpe R29C-conjugated Cy5 in liposomes. Vertical axis is fraction of observations. (**E**) Expected populations of photobleaching events for dimer, trimer, and tetramer channel architectures given a 72% labeling efficiency. A second dataset with BPe T3C gave similar results (**Figure 4—figure supplement 1**). (**F**) Representative measurements of F⁻ efflux from liposomes reconstituted with increasing amounts of Ec2. F⁻ efflux from liposomes loaded with 300 mM KF, as in **Figure 2**. (**G**) Poisson-dump analysis for data in (**F**). Solid curves (**Equation 1**) are determined by calibrating the system with a 104 kDa CLC protein (**Figure 4—figure supplement 2**), with oligomer number indicated (n = 1–4). Similar results were obtained with Bpe (**Figure 4—figure supplement 3**).

The following figure supplements are available for figure 4:

**Figure supplement 1**. Single-Fluc photobleaching.

**Figure supplement 2**. Calibration of Poisson-dump experiments.

**Figure supplement 3**. Poisson-dump data for Fluc-Bpe with expected for the monomer, dimer, trimer, and tetramer curves.

channel insertion into liposomes during reconstitution (**Goldberg and Miller, 1991**; **Maduke et al., 1999**; **Walden et al., 2007**); as an increasing amount of protein is reconstituted into a fixed amount of liposomes, the fraction of liposomes devoid of protein, $f_o$, decreases exponentially. This protein-free fraction may be readily determined from F⁻ efflux experiments, as above. Upon Vln addition, F⁻ exits the liposomes that contain functionally active channels, but remains trapped within the protein-free fraction $f_0$, which is determined by detergent addition (**Figure 4F**):

$$f_0 = \exp\left(-\rho/nMk\right), \qquad (1)$$

where $\rho$ is the experimentally varied protein density (moles Fluc protein/mg lipid), M is the molecular weight of the Fluc subunit, n is the number of subunits in the active channel, and k is a constant dependent on the poorly known size and shape distribution of the liposomes. This constant is determined by calibrating the system with a channel of known molecular weight; for calibration we use Cl⁻ efflux mediated by a high-turnover mutant of the Cl⁻/H⁺ antiporter CLC-ec1-E148A/Y445A (**Jayaram et al., 2008**), a homodimer of 52 kDa subunits (**Figure 4—figure supplement 2**). **Figure 4G** shows $f_o$ curves expected for monomer, dimer, trimer, and tetramer architecture, along with experimental points from multiple preps, which coincide well with the dimer curve. The best exponential fit

to the Fluc-Ec2 data corresponds to a molecular weight of 31.7 kDa for the functional channel, in unreasonably good agreement with the predicted size of the homodimer, 31.6 kDa. Similar results were obtained in parallel experiments with Fluc-Bpe (*Figure 4—figure supplement 3*).

## Fluc channels are built by dual-topology architecture

In about half of the bacterial genomes in which it is found, Fluc appears as a single gene, as with the homodimeric channels Ec2 and Bpe (*Figure 5A*). Other prokaryotes carry a pair of homologous Fluc genes arranged in tandem (~25% sequence identity), strongly suggestive of primeval gene duplication. To test the function of paired Fluc proteins, we individually expressed, purified and reconstituted each of the twin Flucs of *Lactobacillus acidophilus*, La1 and La2 (*Figure 1*). Although the low yield of La2 precludes detailed biochemical analysis, functional examination of these fraternal twins produces an unambiguous result (*Figure 5B*). Neither La1 nor La2 alone catalyzes F⁻ efflux from liposomes, but co-reconstituting both proteins produces rapid F⁻ efflux, indicating that the active channel requires heteromeric assembly.

Comparison of Fluc sequences of singletons vs fraternal twins shows striking differences in charge distributions on the loops connecting the helices. The twin Flucs possess abundantly charged loops; for one of the pair, a positive charge bias resides on the N- and C-termini and loop 2, while its homologous partner exhibits a converse bias, with excess positive charge on loops 1 and 3 (*Figure 5D*). Thus, according to the positive-inside rule for charge distribution in bacterial membrane proteins, (*von Heijne, 1989*) one of the twins is predicted to insert into the membrane with the termini in the cytoplasm, and the second should be oppositely oriented. In striking contrast, the singletons display relatively balanced charge distributions on either face. In a genomic analysis of small membrane proteins, including Fluc, Rapp and colleagues (*Rapp et al., 2006*) proposed that a balanced Arg+Lys distribution in loops indicates dual-topology oligomeric construction, in which subunits insert randomly in both transmembrane orientations, and those of opposite orientations associate together

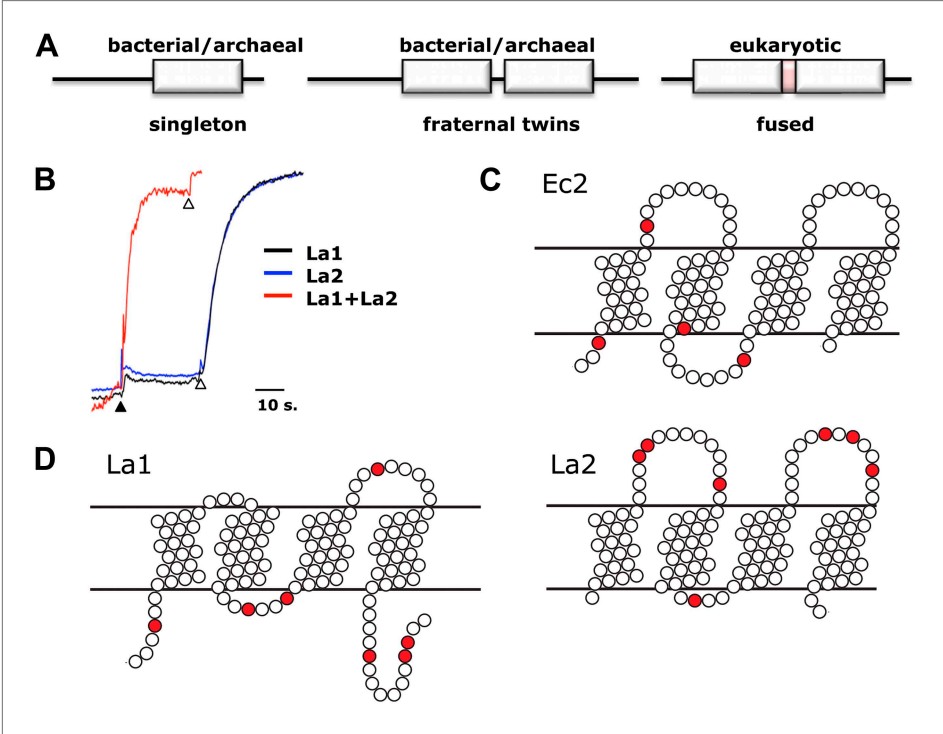

**Figure 5**. Assembly and architecture of Fluc channels. (**A**) Arrangement of Fluc genes in bacterial and eukaryotic genomes. (**B**) Heterodimeric assembly of homologous Fluc subunits. F⁻ efflux was followed by the light-scattering method from liposomes reconstituted with La1 only (40 pmol/mg lipid, black), La2 (40 pmol protein/mg, blue), or both (20 pmol each/mg lipid, red). (**C** and **D**) Sequence-based evidence for dual-topology assembly. Distribution of arginines and lysines (red circles) on loops for Ec2, a singleton-type Fluc or for La1 and La2, homologous-pair Flucs.

(*Figure 5C*). Do singletons like Ec2 and Bpe find partners in the membrane from a pool of randomly oriented Flucs to form inverted-topology homodimers? Do the fraternal Flucs find their homologous companions oppositely oriented in the membrane? A strong indication of antiparallel assembly appears in eukaryotic Fluc genomes. These genes all code for a pair of homologous Fluc sequences within a single open reading frame, and the linker connecting them contains a predicted TM helix (*Figure 5A*). This 'inversion linker' would force the second Fluc domain to adopt a transmembrane topology upside down with respect to the first. These genomic characteristics of Fluc channels constitute compelling evidence for dual-topology construction, but since no previously known ion channels are built in this way, experimental support is needed to test such sequence-based suggestions.

We approach the question in two ways. First, amino-group cross-linking of Fluc subunits is examined in detergent micelles. Even though the Ec2 subunit contains four amino groups, the dimer cannot be cross-linked by glutaraldehyde (*Figure 6A*). Since these amines are all located on the same face of the protein—in loop 2 and both termini—glutaraldehyde would be expected to cross-link a conventional parallel dimer but not an antiparallel dimer. With an additional lysine substituted on the opposite face, in loop 1 (R25K) or loop 3 (N95K), glutaraldehyde treatment now leads to the appearance of a dimer band on the PAGE gel (*Figure 6A*). Similarly, lysine-less Bpe is not cross-linked by glutaraldehyde, despite its solvent-accessible N-terminal amino group. Single lysine mutations on the side opposite to the N-terminus, in loop 1 (R29K) or loop 3 (R95K), beget a cross-linked dimer (*Figure 6B*), whereas solvent-accessible lysines on loops 2 (R68K), on the C-terminus (R130K), or on both do not (*Figure 6B*). These results are in natural harmony with antiparallel orientation and are difficult to understand in terms of a parallel-topology dimer, but do not by themselves provide rigorous proof.

A second test of the dual-topology idea applies the fused-dimer strategy of Schuldiner and colleagues (*Steiner-Mordoch et al., 2008*; *Nasie et al., 2010*) to the Fluc La1/La2 heterodimer. We expressed several constructs in which La1 and La2 are fused together with linkers designed to force the two domains into parallel or antiparallel orientations (*Figure 6C*). For the antiparallel construct, Laf-TM, the domains are linked with a non-dimerizing glycophorin-A TM helix (*Lemmon et al., 1992*; *Fleming et al., 1997*). Guided by eukaryotic Fluc sequences, we made loop 4 long and hydrophilic (18 residues), and loop 5 short (8 residues). The domains of the parallel constructs, LapA

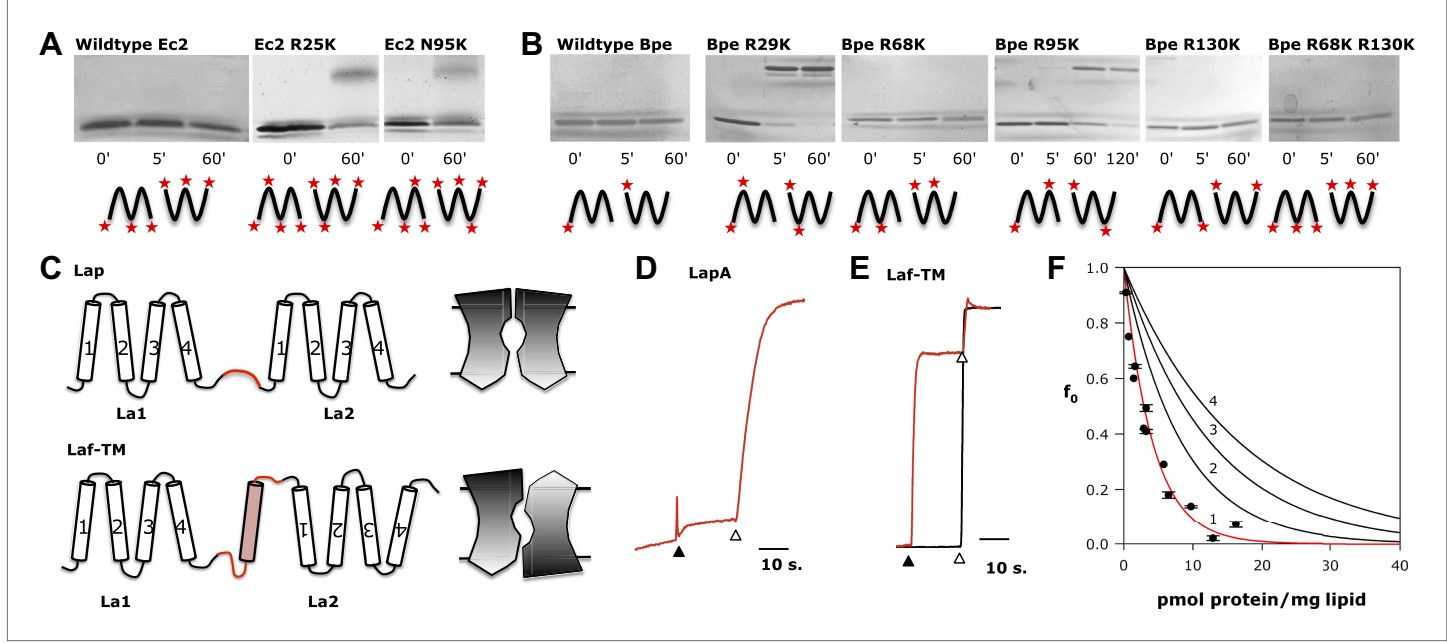

**Figure 6**. Fluc architecture. (**A** and **B**) Crosslinking patterns. 15% SDS-PAGE of WT Ec2 (**A**) or Bpe (**B**) and indicated lysine mutants treated with 0.125% glutaraldehyde for the indicated times. Locations of all primary amino groups predicted for antiparallel dimers are indicated (stars). (**C**) Design of engineered constructs that force La1 and La2 into parallel or antiparallel orientation, with icons envisioning pore-symmetry in each case. (**D**) F⁻ efflux from liposomes for parallel construct LapA (7.5 pmol/mg). (**E**) F⁻ and Cl⁻ efflux from liposomes for antiparallel construct Laf-TM (7.5 pmol/mg). (**F**) Poisson-dump analysis for Laf-TM. Predicted curves are as in *Figure 4*.

and LapB, are connected by hydrophilic linkers of 14 or 26 residues, respectively. All three fused constructs were expressed, purified, and reconstituted into liposomes, Laf-TM giving the best yield and chromatographic monodispersity (*Figure 1—figure supplement 2*). Both parallel constructs have minimal transport activity, as shown for LapA (*Figure 6D*). In marked contrast, Laf-TM behaves quantitatively in Poisson-dump experiments like a fully active monomeric F⁻ channel (*Figure 6E,F*) with high selectivity for F⁻ over Cl⁻. This result strongly implies that functional Fluc channels are built with the two 4-TM subunits or domains in antiparallel transmembrane topology, an arrangement that prohibits axial symmetry along the pore (*Figure 6C*).

## Discussion

In this work, we have taken four initial steps towards understanding the molecular character of the Fluc family of F⁻ exporters. The results demonstrate that the mechanism by which Fluc transports F⁻ is thermodynamically passive electrodiffusion through a transmembrane channel, that the channel is unusually F⁻-selective, and that it is constructed as a dimer of subunits or homologous domains. In addition, the two subunits in the functional dimer are very likely oriented antiparallel to each other within the membrane.

How could a passive channel move F⁻ out of the cell against an environmental F⁻ challenge? It might seem that an active transport mechanism would be required, as in bacteria that use $CLC^F$–type F⁻/H⁺ antiporters. However, Fluc channels can work to effectively expel F⁻ in two ways. First, the negative membrane potential maintained by many classes of metabolizing cells tilts the anion's equilibrium towards expulsion. Second, the high $pK_a$ of HF (~3.4), anomalous among haloacids, dictates that in weakly acidic environments, a significant amount of extracellular HF is present (*Baker et al., 2012*). This membrane-permeant acid readily enters the cell and dissociates at the higher cytoplasmic pH. F⁻ would therefore accumulate far above its extracellular concentration if no conductive pathway for the anion were present in the membrane, but a passive F⁻ channel would undermine this weak-acid accumulation effect.

A deeper enigma is Fluc's remarkable discrimination between F⁻, which is so difficult to desolvate, and Cl⁻, which is orders of magnitude more abundant in the environment. Liposome flux experiments establish a far more dramatic F⁻/Cl⁻ selectivity than the 100-fold lower limit set by electrical recording. Anion-efflux assays measure Cl⁻ turnover <30 ions/s, while in the same assays with F⁻, unitary turnover rates exceed 30,000 s⁻¹, and direct single-channel recording yields a turnover of $10^6$–$10^7$ F⁻/sec. These values conservatively put F⁻/Cl⁻ selectivity >$10^4$-fold, the highest value known between close analogs for any ion channel. It is this formidable selectivity that allows a bacterium or yeast to express a constitutively open F⁻ channel in its energy-coupling membrane while avoiding a massive Cl⁻ conductance that would otherwise catastrophically collapse its membrane potential. High ion selectivity requires that F⁻ be largely desolvated as it passes through the Fluc channel (*Hille, 2001*), an energetically demanding step (>100 kcal/mol) that has been an obstacle to the development of F⁻-selective small-molecule ligands (*Cametti and Rissanen, 2009*). This chemical problem motivates future work to identify selectivity regions in Fluc sequences and, ultimately, structures.

The molecular architecture inferred here is unconventional and unexpected. Most ion channel proteins are built on a barrel-stave plan in which three to seven similar subunits surround an axial pore running perpendicular to the membrane plane, the wider pores requiring a larger number of subunits. But an ion-conduction pore formed by a two-stave barrel—a dimer—is unprecedented until now. We are obliged to point out, however, that Fluc's high open probability precludes a definitive conclusion that the observed single-channel currents are mediated by a single pore rather than two in parallel, especially in light of the infrequent subconductance (*Figure 3*). However, this substate is not precisely half the conductance of the fully open channel, and the protein's small size makes a two-pore dimer, along the lines of CLC channels (*Middleton et al., 1994*; *Ludewig et al., 1996*; *Middleton et al., 1996*), unlikely, but a picture like this is not firmly ruled out at our investigation's early stage.

A final surprise that provides an intriguing glimpse into the evolution of membrane transport proteins is the dual-topology construction inferred for Fluc. Unprecedented among known ion channels, dual topology has been observed in the SMR family of proton-coupled multidrug antiporters such as EmrE (*Schuldiner, 2009, 2012*; *Morrison et al., 2012*), and it recalls the inverted structural repeats that provide the mechanistic scaffolds of many transporters (*Forrest et al., 2010*). Oppositely oriented repeats within subunits are rare in ion channels, and so it is unclear whether this architecture provides particular mechanistic advantages for Flucs. The inverted structural repeats of transporters

almost certainly evolved from gene duplication and subsequent fusion of sequences coding for dimeric proteins with an odd number of TM helices (*Zuckerkandl and Pauling, 1965*; *Forrest and Rudnick, 2009*). The Fluc family obligingly provides examples in modern genomes of all stages along such an evolutionary pathway, including duplicated gene-pairs in bacteria and fused homologues in eukaryotes. Fusion of the latter genes must have required evolutionary gymnastics to capture a TM helix as an inversion linker for maintaining antiparallel topology of subunits with an even number of membrane crossings.

Finally, we note a conspicuous analogy to Fluc construction: the aquaporin channel. This protein's narrow, water-permeable pore is formed within a single 6-TM subunit on the interface between two structurally similar domains in inverted orientation with respect to each other (*Harries et al., 2004*). It is thus entirely feasible to construct a narrow pore at a dual-topology dimer interface. The Fluc-aquaporin analogy is also of interest since the desolvated $F^-$ ion, with a radius of 1.3 Å, is almost the same size as $H_2O$.

## Materials and methods

### Reagents

Chemicals from Sigma-Aldrich (St. Louis, MO) were of highest grade obtainable. *E. coli* mixed phospholipids (EPL), 1-palmitoyl, 2-oleoylphosphatidylethanolamine (POPE), 1-palmitoyl, 2-oleoyl-phosphatidylcholine (POPC), and 1-palmitoyl, 2-oleoylphosphatidylglycerol (POPG) were obtained from Avanti Polar Lipids, detergents from Anatrace, and fluorophores from Invitrogen or GE-Biosciences. K-isethionate solutions were prepared from isethionic acid (Wako Chemicals, Richmond, VA) titrated with KOH.

### Design of Fluc constructs

Constructs used in this study are summarized in *Table 1*. Synthetic gene constructs for Fluc-Ec2, Fluc-Bpe, Fluc-La1, and Fluc-La2 were inserted into a pASK vector with a C-terminal LysC recognition site and hexahistidine tag (TRKAASLVPRGSGGHHHHHH) (*Maduke et al., 1999*). The fused constructs Laf-TM, LapA, and LapB were formed from the sequence of La1 followed by a linker sequence (*Table 1*) leading into the sequence of La2 (without the first methionine), to which a C-terminal hexahistidine tag was appended. Laf-TM contained a transmembrane linker composed of a non-dimerizing glycophorin A helix (*Lemmon et al., 1992*). Site-directed mutagenesis was performed using standard PCR techniques. All mutants were confirmed functionally active in a liposome-based $F^-$ efflux assay.

### Protein purification and liposome preparation

*E. coli* (BL21-DE3) transformed with the pASK-IBA2 plasmid bearing Fluc constructs was induced with anhydrotetracycline at an optical density of 0.5–1.0, and protein was expressed for 1 hr, 37°C. Cells were lysed by sonication and extracted with 60 mM decylmaltoside (DM) for 2 hr at 4°C. After pelleting the cell debris, the supernatant was passed over cobalt affinity beads (Talon, 1 ml/l culture), washed with 100 mM NaCl, 45 mM imidazole, and eluted with 300 mM imidazole. The eluate was diluted 10-fold into ion-exchange (IE) buffer, 10 mM NaCl, 5 mM DM, 10 mM 2-(N-morpholino)ethanesulfonic acid (MES)-NaOH pH 6.5, and applied to a 0.5-ml cation-exchange column (Poros 50 HS). After washing with 10 vol of IE buffer, protein was eluted with IE buffer + 400 mM NaCl and was further purified on a Superdex 200 size-exclusion column (SEC) in 100 mM NaCl, 10 mM NaF, 10 mM 4-(2-hydroxyethyl)-1-piperazineethanesulfonic acid [HEPES]), pH 7, 5 mM DM. Typical protein yields were 50–150 µg/l culture for Bpe, Ec2, and La1, ~30 µg/l for the fused La constructs, and ~5 µg/l for La2. Proteoliposomes were formed by dialysis of a micellar mix of protein (0.02–2 µg protein/mg lipid), lipid (10–25 mg/ml), and 35 mM 3-[(3-Cholamidopropyl)dimethylammonio]-1-propanesulfonate (CHAPS) against the desired intraliposomal solution at room temperature for 36 hr. Liposomes were stored in aliquots at −80°C until use. For cysteine labeling applications, the protein was prepared in the presence of 10 mM Tris (2-carboxyethyl) phosphine (TCEP; Toronto Research Chemicals, Toronto, Canada) until it was removed at the size exclusion step.

### Topology determination

10 mM stocks of AlexaFluor 647-C2-maleimide and fluorescein succinimidyl ester were prepared in anhydrous DMSO and kept in the dark at −80°C until use. The Bpe construct contained unique cysteine mutations T3C, N31C, or E94C for maleimide labeling, with an R29K background to increase labeling by the succinimidyl ester. Proteoliposomes (POPC/POPG, 20 mg/ml) were prepared with 2 µg

Bpe/mg lipid, with an intraliposomal solution of 300 mM NaF, 25 mM HEPES pH 7.5, and 1 mM cysteine. A 100-µl sample of the liposomes was treated with LysC (0.05 U, 1 hr, 22°C) to completely cleave all externally exposed C-terminal His tags and was then centrifuged through a 1.5-ml G-50 column equilibrated with 300 mM NaF, 25 mM HEPES pH 7.5 to remove cysteine and LysC from the external solution. AlexaFluor maleimide (50 µM) was then added to label external cysteine residues, and after an hour freshly prepared phenylmethylsulfonyl fluoride (PMSF) was added to quench residual LysC, and 10 mM cysteine to quench the maleimide. Liposomes were disrupted by 150 mM β-octylglucoside, and fluorescein succinimidyl ester (50 µM) was added for 1 hr to stain protein amino groups before quenching with 10 mM Tris. Samples were run on 10–20% SDS-PAGE gels and fluorescent bands were visualized using a Typhoon 9410 Variable Mode Imager (GE Healthcare). Cysteine-conjugated protein was visualized with the red laser (633 nm excitation, 670 nm emission), and fluorescein-conjugated total protein with the blue laser (488 nm excitation, 526 nm emission). Fluorescein was used for staining total protein because of the large amount of lipid in the sample, which interferes with Coomassie or silver stain.

### Anion transport assays

Anion efflux assays were performed as described previously (*Stockbridge et al., 2012*). In short, liposomes prepared with 10 mg/ml *E. coli* polar lipids containing 150 mM KF, 150 mM KCl, and 25 mM HEPES pH 7 were extruded 21 times through a 400-nm membrane filter and passed over a 1.5-ml G-50 Sephadex column equilibrated in external buffer composed of 300 K-isethionate, 25 mM HEPES pH 7, 1 mM KF or KCl, according to the ion measured. Liposomes were diluted 20-fold into external buffer in a stirred cuvette, and efflux was initiated by addition of 1 µM $K^+$ ionophore valinomycin. After ~30 s, 50 mM β-octylglucoside was added to disrupt the liposomes. $Cl^-$ or $F^-$ appearance in the external solution was monitored using homemade Ag/AgCl or Cole-Parmer $LaF_3/EuF_3$ electrodes, respectively; under the ionic conditions used here, these electrodes are ideally selective for $Cl^-$ or $F^-$, and they show no cross-reactivity towards the other anion.

In some experiments, $F^-$ efflux was monitored by 90° light scattering at 600 nm in a fluorimeter. A liposome sample containing 300 mM KF, 25 mM HEPES-KOH pH 7 was diluted 200-fold into 2 ml of a degassed isotonic solution containing 300 mM K- or Na-isethionate, 25 mM HEPES-KOH pH 7 in a stirred cuvette, and efflux was initiated by 1 µM Vln. Water efflux maintaining osmotic balance leads to time-dependent shrinking and flattening of the initially spherical liposomes, accompanied by a ~10% increase in 90° light scattering (*Jin et al., 1999*; *Stockbridge et al., 2012*). After transport was complete, p-trifluoromethoxyphenyl hydrazine (FCCP) was added to the cuvette, effectively making the protein-free liposomes specifically permeable to $F^-$ to provide a quantitative measurement of the fraction of liposomes, $f_0$, free of Fluc channels. For Poisson-dump experiments, data were collected for proteins from four independent preparations of CLC-ec1 and Fluc-Ec2, and two independent preparations of Fluc-Bpe, Fluc-Laf, Fluc-LapA, and Fluc-LapB.

### Planar lipid bilayer recording

Planar bilayer recording was as previously described (*Accardi et al., 2004*), using a POPE/POPG phospholipid mixture to form the bilayer and 150 mM NaF-1.5% agar salt bridges to connect the recording chambers to the Ag/AgCl electrodes through 1 M KCl wells. Liposomes (0.5 µl added to the cis solution) were fused into the bilayer, with 300 mM or 30 mM NaF, 15 mM Mops-NaOH pH 7, in the cis or trans chamber respectively, trans defined as electrical ground. Ionic conditions for recording were established by perfusion of either chamber. Macroscopic I–V relations were determined with families of voltage pulses (0.5–1 s) from a holding potential of zero to command-voltages from −100 to +150 mV in 10-mV increments. Current output from a Axopatch 200A amplifier was low-pass filtered at 500 Hz and sampled at 2 kHz in pCLAMP software. Voltages were corrected for junction potential, <3 mV. Signals for single-channel analysis were subsequently filtered digitally at 100 Hz.

### SEC-SLS-UV-RI

The molecular mass of Fluc in detergent was determined using the static light scattering/refractive index method (*Folta-Stogniew, 2006*; *Slotboom et al., 2008*) implemented at the W. M. Keck Foundation Biotechnology Resource Laboratory, Yale University. Briefly, purified protein was passed over a Superdex-75 SEC column equilibrated in 400 mM NaF, 10 mM MES pH 6.5, 5 mM DM and coupled in-line with light scattering, refractive index, and UV detectors. Ovalbumin, transferase,

bovine serum albumin, and carbonic anhydrase were used as standards. The molecular mass was determined in two ways: by the three-detector method (*Hayashi et al., 1989*), and from light scattering intensity at several angles (ASTRA software, Wyatt Technology Corp.).

## Single-molecule TIRF photobleaching

For the labeling reaction, freshly prepared Bpe with a single cysteine substitution (T3C or R29C) was incubated with a 20-fold molar excess of Cy5-maleimide for 1 hr at room temperature in the dark and quenched with a 100-fold molar excess of cysteine. To wash away excess fluorophore, the protein was bound to cobalt affinity beads and washed with 45 column volumes of cobalt wash buffer. The protein was eluted with 400 mM imidazole and the labeling efficiency was calculated from the protein absorbance at 280 nm ($\varepsilon_{Bpe}$ = 38,960 $M^{-1}$ $cm^{-1}$, calculated from protein sequence), and the Cy5 absorbance at 655 nm ($\varepsilon_{Cy5}$ = 2.5 × $10^5$ $M^{-1}$ $cm^{-1}$), with spillover correction of 0.017. Parallel experiments with wildtype, cysteine-free Bpe showed that the background labeling was <2% under the same conditions. Protein was reconstituted into liposomes at very low density (0.15 µg/mg lipid).

Flow chambers were prepared using two glass coverslips that were cleaned by sonication in 2% micro-90 detergent (Cole-Parmer, Vernon Hills, IL), ethanol and 0.1 M KOH. Liposomes were formed immediately prior to analysis. BPe T3C liposomes were sonicated vigorously for 10 min until clarity, yielding ~30 nm liposomes. BPe R29C liposomes were extruded 21 times through a 100 nm filter, then 21 times through 30 nm filter (Avestin, Ottawa, Canada) to form small vesicles. Liposomes were diluted ~$10^5$-fold in reconstitution buffer, and loaded into the flow chamber, where they spontaneously adhere to the glass surface. The fluorescence from each liposome was measured using a custom through-objective TIRF microscope built for single molecule detection (*Friedman et al., 2006*). Cy5 fluorescence was measured by illuminating the sample with a helium-neon 633 nm laser (75 µW power) and collecting the emitted light through a 633 nm long-pass filter (1 s per frame). Each spot was centered inside a 3 × 3 pixel area using linear interpolation and drift-correction, and the intensity was integrated over the duration of the experiment.

## Cross-linking

For glutaraldehyde cross-linking experiments, ~0.1 mg/ml freshly purified protein in 20 mM 3-(N-morpholino)propanesulfonic (MOPS), pH 7.5, 100 mM NaCl, 10 mM NaF, and 5 mM DM was incubated with 0.125% glutaraldehyde (~500-fold molar excess) for 5–120 min. The reaction was quenched with a 10-fold molar excess of Tris-HCl, pH 7.5, and samples were run on a 15% SDS-PAGE gel.

## Acknowledgements

We are grateful to Jeff Gelles and Larry Friedman for use of the TIRF microscope, and to Ewa Folta-Stogniew for her rigorous implementation of the SEC-LS/UV/RI experiments (WM Keck facility, Yale University, supported by NIH Award Number 1S10RR023748-01). We also thank Ashley Brammer for setting up the F⁻ electrode system and Phill Stansfeld for provocative insights about the transmembrane topology of Fluc proteins.

## Additional information

### Funding

| Funder | Grant reference number | Author |
| --- | --- | --- |
| Howard Hughes Medical Institute | | Christopher Miller |
| National Institutes of Health | K99-GM101016 | Janice L Robertson |

The funders had no role in study design, data collection and interpretation, or the decision to submit the work for publication.

### Author contributions

RBS, JLR, CM, Conception and design, Acquisition of data, Analysis and interpretation of data, Drafting or revising the article; LK-P, Acquisition of data

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
