## [Decision Letter]

Thank you for sending your work entitled “A family of fluoride-specific ion channels with dual-topology architecture” for consideration at *eLife*. Your article has been favorably evaluated by a Senior editor and 3 reviewers, one of whom is a member of our Board of Reviewing Editors.

The following individuals responsible for the peer review of your submission have agreed to reveal their identity: Richard Aldrich (Reviewing editor), Olaf Andersen, and Joe Mindell (peer reviewers).

The Reviewing editor and the other reviewers discussed their comments before we reached this decision. This interesting and novel paper from the Miller lab reports the discovery and characterization of a new class of fluoride-conducting ion channels. These channels, referred to as “Fluc”, were found as a family of membrane proteins under the control of an F^-^-sensing riboswitch. Here, the Miller lab expressed and purified several homologs, and find that they mediate dissipative and highly selective F^-^ fluxes in liposomes and lipid bilayer membranes. Single channel events present themselves when liposomes containing low concentrations of Fluc proteins are fused to lipid bilayers, conclusively revealing the functional classification of these proteins. Simple labeling experiments sketch out a likely transmembrane topology. Several lines of evidence point to a dimeric functional unit, including crosslinking, “poisson-dump” assays, and single-molecule photobleaching. Surprisingly, the protein appears to form antiparallel dimers, as has been recently described for the multidrug transporter EmrE but for few, if any, other native membrane proteins. The work is thorough and rigorous, and the arguments are strong and convincing.

The manuscript is clearly written and provides important new insights into cellular F^-^ homeostasis, as it provides solid experimental evidence that the Fluc family of fluoride channels, with their impressive selectivity for F^-^ over Cl^-^, enable cells to maintain a low intercellular F^-^ with no direct expenditure of metabolic energy.

[Editors’ note: minor issues and corrections have not been included, so there is not an accompanying Author response.]